# Monte Carlo Augmented Actor-Critic for Sparse Reward Deep Reinforcement Learning from Suboptimal Demonstrations

**Albert Wilcox**
UC Berkeley
albertwilcox@berkeley.edu

**Ashwin Balakrishna**
UC Berkeley
ashwin_balakrishna@berkeley.edu

**Jules Dedieu**
UC Berkeley
jules_dedieu@berkeley.edu

**Wyame Benslimane**
UC Berkeley
wyame.benslimane@berkeley.edu

**Daniel S. Brown**
University of Utah
dsbrown@cs.utah.edu

**Ken Goldberg**
UC Berkeley
goldberg@berkeley.edu

## Abstract

Providing densely shaped reward functions for RL algorithms is often exceedingly challenging, motivating the development of RL algorithms that can learn from easier-to-specify sparse reward functions. This sparsity poses new exploration challenges. One common way to address this problem is using demonstrations to provide initial signal about regions of the state space with high rewards. However, prior RL from demonstrations algorithms introduce significant complexity and many hyperparameters, making them hard to implement and tune. We introduce Monte Carlo augmented Actor-Critic (MCAC), a parameter free modification to standard actor-critic algorithms which initializes the replay buffer with demonstrations and computes a modified $Q$-value by taking the maximum of the standard temporal distance (TD) target and a Monte Carlo estimate of the reward-to-go. This encourages exploration in the neighborhood of high-performing trajectories by encouraging high $Q$-values in corresponding regions of the state space. Experiments across 5 continuous control domains suggest that MCAC can be used to significantly increase learning efficiency across 6 commonly used RL and RL-from-demonstrations algorithms. See `https://sites.google.com/view/mcac-rl` for code and supplementary material.

## 1   Introduction

Reinforcement learning has been successful in learning complex skills in many environments [Mnih et al., 2015, Silver et al., 2016, Schulman et al., 2017], but providing dense, informative reward functions for RL agents is often very challenging [Xie et al., 2019, Krakovna et al., 2020, Wu et al., 2020]. This is particularly challenging for high-dimensional control tasks, in which there may be a large number of factors that influence the agent's objective. In many settings, it may be much easier to provide sparse reward signals that simply convey high-level information about task progress, such as whether an agent has completed a task or has violated a constraint. However, optimizing RL policies given such reward signals can be exceedingly challenging, as sparse reward functions may not be able to meaningfully distinguish between a wide range of different policies.

36th Conference on Neural Information Processing Systems (NeurIPS 2022).

This issue can be mitigated by leveraging demonstrations, which provide initial signal about desired behaviors. Though demonstrations may often be suboptimal in practice, they should still serve to encourage exploration in promising regions of the state space, while allowing the agent to explore behaviors which may outperform the demonstrations. Prior work has considered a number of ways to leverage demonstrations to improve learning efficiency for reinforcement learning, including by initializing the policy to match the behavior of the demonstrator [Rajeswaran et al., 2018, Peng et al., 2019], using demonstrations to explicitly constrain agent exploration [Thananjeyan et al., 2020, 2021, Wilcox et al., 2021], and introducing auxiliary losses to incorporate demonstration data into policy updates [Nair et al., 2018, Hester et al., 2018, Gao et al., 2018]. While these algorithms have shown impressive performance in improving the sample efficiency of RL algorithms, they add significant complexity and hyperparameters, making them difficult to implement and tune for different tasks.

We present Monte Carlo augmented Actor-Critic (MCAC), which introduces an easy-to-implement, but highly effective, change that can be readily applied to existing actor-critic algorithms without the introduction of any additional hyperparameters and only minimal additional complexity. The idea is to encourage initial optimism in the neighborhood of successful trajectories, and progressively reduce this optimism during learning so that it can continue to explore new behaviors. To operationalize this idea, MCAC introduces two modifications to existing actor-critic algorithms. First, MCAC initializes the replay buffer with task demonstrations. Second, MCAC computes a modified target $Q$-value for critic updates by taking the maximum of the standard temporal distance targets used in existing actor critic algorithms and a Monte Carlo estimate of the reward-to-go. The intuition is that Monte Carlo value estimates can more effectively capture longer-term reward information, making it possible to rapidly propagate reward information from demonstrations through the learned $Q$-function. This makes it possible to prevent underestimation of values in high-performing trajectories early in learning, as high rewards obtained later in a trajectory may be difficult to initially propagate back to earlier states with purely temporal distance targets [Wright et al., 2013]. Experiments on five continuous control domains suggest that MCAC is able to substantially accelerate exploration for both standard RL algorithms and recent RL from demonstrations algorithms in sparse reward tasks.

## 2   Related Work

### 2.1   Reinforcement Learning from Demonstrations

One standard approach for using demonstrations for RL first uses imitation learning [Argall et al., 2009] to pre-train a policy, and then fine-tunes this policy with on-policy reinforcement learning algorithms [Schaal et al., 1997, Kober and Peters, 2014, Peng et al., 2019, Rajeswaran et al., 2018]. However, initializing with suboptimal demonstrations can hinder learning and using demonstrations to initialize only a policy is inefficient, since they can also be used for $Q$-value estimation.

Other approaches leverage demonstrations to explicitly constrain agent exploration. Thananjeyan et al. [2020] and Wilcox et al. [2021] propose model-based RL approaches that uses suboptimal demonstrations to iteratively improve performance by ensuring consistent task completion during learning. Similarly, Jing et al. [2020] also uses suboptimal demonstrations to formulate a soft-constraint on exploration. However, a challenge with these approaches is that they introduce substantial algorithm complexity, making it difficult to tune and utilize these algorithms in practice. For example, while Thananjeyan et al. [2020] does enable iterative improvement upon suboptimal demonstrations, they require learning a model of system dynamics and a density estimator to capture the support of successful trajectories making it challenging to scale to high-dimensional observations.

Finally, many methods introduce auxiliary losses to incorporate demonstrations into policy updates [Kim et al., 2013, Gao et al., 2018, Kang et al., 2018]. Deep Deterministic Policy Gradients from Demonstrations (DDPGfD) [Vecerik et al., 2018] maintains all the demonstrations in a separate replay buffer and uses prioritized replay to allow reward information to propagate more efficiently. Nair et al. [2018], and Hester et al. [2018] use similar approaches, where demonstrations are maintained separately from the standard replay buffer and additional policy losses encourage imitating the behavior in the demonstrations. Meanwhile, RL algorithms such as AWAC [Nair et al., 2021] pretrain using demonstration data to constrain the distribution of actions selected during online exploration. While these methods often work well in practice, they often increase algorithmic complexity and introduce several additional hyperparameters that are difficult and time consuming to tune. By

contrast, MCAC does not increase algorithmic complexity, is parameter-free, and can easily be wrapped around any existing actor-critic algorithm.

## 2.2 Improving $Q$-Value Estimates

The core contribution of this work is an easy-to-implement, yet highly effective, method for stabilizing actor-critic methods for sparse reward tasks using demonstrations and an augmented $Q$-value target. There has been substantial literature investigating learning stability challenges in off-policy deep $Q$-learning and actor-critic algorithms. See Van Hasselt et al. [2018] for a more thorough treatment of the learning stability challenges introduced by combining function approximation, bootstrapping, and off-policy learning, as well as prior work focused on mitigating these issues.

One class of approaches focuses on developing new ways to compute target $Q$-values [Konidaris et al., 2011, Wright et al., 2013, Van Hasselt et al., 2016, Schulman et al., 2016]. Van Hasselt et al. [2016] computes target $Q$-values with two $Q$-networks, using one to select actions and the other to measure the value of selected actions, which helps to prevent the $Q$-value over-estimation commonly observed in practice in many practical applications of $Q$-learning. TD3 [Fujimoto et al., 2018] attempts to address overestimation errors by taking the minimum of two separate $Q$-value estimates, but this can result in underestimation of the true $Q$-value target. Kuznetsov et al. [2020] uses an ensemble of critics to adaptively address estimation errors in $Q$-value targets, but introduces a number of hyperparameters which must be tuned separately for different tasks. Bellemare et al. [2016] and Ostrovski et al. [2017] consider a linear combination of a TD-1 target and Monte Carlo target. Wright et al. [2015], Schulman et al. [2016] and Sharma et al. [2018] consider a number of different estimators of a policy's value via $n$-step returns, which compute $Q$-targets using trajectories with $n$ contiguous transitions followed by a terminal evaluation of the $Q$-value after $n$ steps. Each of these targets make different bias and variance tradeoffs that can affect learning dynamics.

Similar to our work, Wright et al. [2013] explore the idea of taking a maximum over a bootstrapped target $Q$-value (TD-1 target) and a Monte Carlo estimate of the return-to-go to improve fitted $Q$-iteration. However, Wright et al. [2013] focuses on fully offline $Q$-learning and only considers simple low-dimensional control tasks using $Q$-learning with linear value approximation. There are numerous reasons to extend these ideas to online deep RL. First, deep RL algorithms are often unstable, and using the ideas in Wright et al. [2013] to improve $Q$ estimates is a promising way to alleviate this, as we empirically verify. Second, while offline learning has many important applications, online RL is far more widely studied, and we believe it is useful to study the effects these ideas have in this setting. To the best of our knowledge, MCAC is the first application of these ideas to online actor-critic algorithms with deep function approximation, and we find that it yields surprising improvements in RL performance on complex high-dimensional continuous control tasks.

# 3 Problem Statement

We consider a Markov Decision Process (MDP) described by a tuple $(\mathcal{S}, \mathcal{A}, p, r, \gamma, T)$ with a state set $\mathcal{S}$, an action set $\mathcal{A}$, a transition probability function $p : \mathcal{S} \times \mathcal{A} \times \mathcal{S} \rightarrow [0, 1]$, a reward function $r : \mathcal{S} \times \mathcal{A} \rightarrow \mathbb{R}$, a discount factor $\gamma$, and finite time horizon $T$. In each state $s_t \in \mathcal{S}$ the agent chooses an action $a_t \in \mathcal{A}$ and observes the next state $s_{t+1} \sim p(\cdot|s_t, a_t)$ and a reward $r(s_t, a_t) \in \mathbb{R}$. The agent acts according to a policy $\pi$, which induces a probability distribution over $\mathcal{A}$ given the state, $\pi(a_t|s_t)$. The agent's goal is to find the policy $\pi^*$ which at any given $s_t \in \mathcal{S}$ maximizes the expected discounted sum of rewards,

$$\pi^* = \arg\max_{\pi} \mathbb{E}_{\tau \sim \pi} \left[ \sum_{t=0}^{T} \gamma^t r(s_t, a_t) \right], \tag{3.1}$$

where $\tau = (s_0, a_0, s_1, a_1, \dots s_T)$ and $\tau \sim \pi$ indicates the distribution of trajectories induced by evaluating policy $\pi$ in the MDP.

We make additional assumptions specific to the class of problems we study. First, we assume that all transitions in the replay buffer are elements of complete trajectories; this is a reasonable assumption as long as all transitions are collected from rolling out some policy in the MDP. Second, we assume the agent has access to an offline dataset $\mathcal{D}_{\text{offline}}$ of (possibly suboptimal) task demonstrations. Finally, we focus on settings where the reward function is sparse in the sense that most state transitions do not induce a change in reward, causing significant exploration challenges for RL algorithms.

# 4 Preliminaries: Actor-Critic Algorithms

For a given policy $\pi$, its state-action value function $Q^\pi$ is defined as

$$Q^\pi(s_t, a_t) = \mathbb{E}_{\tau \sim \pi} \left[ \sum_{k=t}^{T} \gamma^{k-t} r(s_k, a_k) \right]. \tag{4.1}$$

Actor-critic algorithms learn a sample-based approximation to $Q^\pi$, denoted $Q_\theta^\pi$, and a policy $\pi_\phi$ which selects actions to maximize $Q_\theta^\pi$, with a function approximator (typically a neural network) parameterized by $\theta$ and $\phi$ respectively. During the learning process, they alternate between regressing $Q_\theta^\pi$ to predict $Q^\pi$ and optimizing $\pi_\phi$ to select actions with high values under $Q_\theta^\pi$.

Exactly computing $Q^\pi$ targets to train $Q_\theta^\pi$ is typically intractable for arbitrary policies in continuous MDPs, motivating other methods for estimating them. One such method is to simply collect trajectories $(s_t, a_t, r_t, s_{t+1}, \ldots s_{T-1}, a_{T-1}, r_{T-1}, s_T)$ by executing the learned policy $\pi_\phi$ from state $s_t$, computing a Monte Carlo estimate of the reward-to-go defined as follows:

$$Q_{\text{MC}}^{\text{target}}(s_t, a_t) = \sum_{k=t}^{T} \gamma^{k-t} r(s_k, a_k), \tag{4.2}$$

and fitting $Q_\theta^\pi$ to these targets.

However, $Q_{\text{MC}}^{\text{target}}$ can be a high variance estimator of the reward-to-go [Schulman et al., 2016, Sutton and Barto, 2018], motivating the popular one-step temporal difference target (TD-1 target) to help stabilize learning:

$$Q_{\text{TD}}^{\text{target}}(s_t, a_t) = r(s_t, a_t) + \gamma Q_{\theta'}^\pi(s_{t+1}, a_{t+1}), \tag{4.3}$$

where $a_{t+1} \sim \pi_\phi(s_{t+1})$, which is recursively defined based on only a single $(s_t, a_t, s_{t+1}, r_t)$ transition. Here $\theta'$ is the parameters of a lagged target network as in [Van Hasselt et al., 2016]. There has also been interest in computing TD-$n$ targets, which instead sum rewards for $n$ timesteps and then use $Q$-values from step $n+1$ [Wright et al., 2015, Schulman et al., 2016, Sharma et al., 2018].

# 5 Monte Carlo augmented Actor-Critic

## 5.1 MCAC Algorithm

The objective of MCAC is to efficiently convey information about sparse rewards from suboptimal demonstrations to $Q_\theta^\pi$ in order to accelerate policy learning while still maintaining learning stability. To do this, MCAC combines two different methods of computing targets for fitting $Q$-functions to enable efficient value propagation throughout the state-action space while also learning a $Q$-value estimator with low enough variance for stable learning. To operationalize this idea, MCAC defines a new $Q$-function target for training $Q_\theta^\pi$ by taking the maximum of the Monte Carlo $Q$-target (eq 4.2) and the temporal difference $Q$-target (eq 4.3): $\max \left[ Q_{\text{TD}}^{\text{target}}(s_t, a_t), Q_{\text{MC}}^{\text{target}}(s_t, a_t) \right]$.

The idea here is that early in learning, a $Q$-function trained only with temporal difference targets will have very low values throughout the state-action space as it may be very difficult to propagate information about delayed rewards through the temporal difference target for long-horizon tasks with sparse rewards [Van Hasselt et al., 2018]. On the other hand, the Monte Carlo $Q$-target can easily capture long-term rewards, but often dramatically underestimates $Q$-values for poorly performing trajectories [Wright et al., 2013]. Thus, taking a maximum over these two targets serves to initially boost the $Q$-values for transitions near high performing trajectories while limiting the influence of underestimates from the Monte Carlo estimate.

MCAC can also be viewed as a convenient way to balance the bias and variance properties of the Monte Carlo and temporal difference $Q$-targets. The Monte Carlo $Q$-target is well known to be an unbiased estimator of policy return, but have high variance [Wright et al., 2013]. Conversely, temporal difference targets are known to be biased, but have much lower variance [Wright et al., 2013, Van Hasselt et al., 2016]. Thus, as the temporal difference target is typically negatively biased (an underestimate of the true policy return) on successful trajectories early in learning due to the challenge of effective value propagation, computing the maximum of the temporal difference and

---

**Algorithm 1** Monte Carlo augmented Actor-Critic

---

**Require:** Offline dataset $\mathcal{D}_{\text{offline}}$.
**Require:** Total training episodes $N$, batch size $M$, Pretraining Steps $N_p$
**Require:** Episode time horizon $T$.
1: Initialize replay buffer $\mathcal{R} := \mathcal{D}_{\text{offline}}$.
2: Initialize agent $\pi_\phi$ and critic $Q_\theta^\pi$ using data from $\mathcal{D}_{\text{offline}}$.
3: **for** $i \in \{1, \dots, N_p\}$ **do**
4:      Sample $\mathcal{B} \subsetneq \mathcal{R}$ such that $|\mathcal{B}| = M$.
5:      Optimize $Q_\theta^\pi$ on $\mathcal{B}$ to minimize loss in eq (5.4). Optimize policy $\pi_\phi$ to maximize $Q_\theta$.
6: **end for**
7: **for** $i \in \{1, \dots, N\}$ **do**
8:      Initialize episode buffer $\mathcal{E} = \{\}$.
9:      Observe state $s_1^i$.
10:      **for** $j \in \{1, \dots, T\}$ **do**
11:          Sample and execute $a_t^i \sim \pi_\theta(s_j^i)$, observing $s_{j+1}^i, r_j^i$.
12:          $\tau_j^i \leftarrow (s_j^i, a_j^i, s_{j+1}^i, r_j^i)$
13:          $\mathcal{E} \leftarrow \mathcal{E} \cup \{\tau_j^i\}$.
14:          Sample $\mathcal{B} \subsetneq \mathcal{R}$ such that $|\mathcal{B}| = M$.
15:          Optimize $Q_\theta^\pi$ on $\mathcal{B}$ to minimize loss in eq (5.4). Optimize policy $\pi_\phi$ to maximize $Q_\theta$.
16:      **end for**
17:      **for** $\tau_j^i \in \mathcal{E}$ **do**
18:          Compute $Q_{\text{MC-}\infty}^{\text{target}}(s_j^i, a_j^i)$ as in eq (5.2).
19:          $\tau_j^i \leftarrow (s_j^i, a_j^i, s_{j+1}^i, r_j^i, Q_{\text{MC-}\infty}^{\text{target}}(s_j^i, a_j^i))$.
20:      **end for**
21:      $\mathcal{R} \leftarrow \mathcal{R} \cup \mathcal{E}$.
22: **end for**

---

Monte Carlo targets helps push the MCAC target value closer to the true policy return. Conversely, the Monte Carlo targets are typically pessimistic on unsuccessful trajectories, and their high variance makes it difficult for them to generalize sufficiently to distinguish between unsuccessful trajectories that are close to being successful and those that are not. Thus, computing the maximum of the temporal difference and Monte Carlo targets also helps to prevent excessive pessimism in evaluating unsuccessful trajectories. Notably, MCAC does not constrain its $Q$-targets explicitly based on the transitions in the demonstrations, making it possible for the policy to discover higher performing behaviors than those in the demonstrations as demonstrated in Section 6.

### 5.2 MCAC Practical Implementation

MCAC can be implemented as a wrapper around any actor-critic RL algorithm; we consider 6 options in experiments (Section 6). MCAC starts with an offline dataset of suboptimal demonstrations $\mathcal{D}_{\text{offline}}$, which are used to initialize a replay buffer $\mathcal{R}$. Then, during each episode $i$, we collect a full trajectory $\tau^i$, where the $j^{\text{th}}$ transition $(s_j^i, a_j^i, s_{j+1}^i, r_j^i)$ in $\tau^i$ is denoted by $\tau_j^i$.

Next, consider any actor-critic method using a learned $Q$ function approximator $Q_\theta(s_t, a_t)$ that, for a given transition $\tau_j^i = (s_j^i, a_j^i, s_{j+1}^i, r_j^i) \in \tau^i \subsetneq \mathcal{R}$ is updated by minimizing the following loss:

$$J(\theta) = \ell\left(Q_\theta(s_j^i, a_j^i), Q^{\text{target}}(s_j^i, a_j^i)\right), \tag{5.1}$$

where $\ell$ is an arbitrary differentiable loss function and $Q^{\text{target}}$ is the target value for regressing $Q_\theta$. We note that $Q^{\text{target}}$ is defined by the choice of actor-critic method. To implement MCAC, we first calibrate the Monte Carlo targets with temporal difference targets (which provide infinite-horizon $Q$ estimates) by computing the infinite horizon analogue of the Monte Carlo target defined in Equation 4.2, which assumes the last observed reward value will repeat forever and uses this to add an infinite sum of discounted rewards, and is given by

$$Q_{\text{MC-}\infty}^{\text{target}}(s_j^i, a_j^i) = \gamma^{T-j+1} \frac{r_T^i}{1-\gamma} + \sum_{k=j}^{T} \gamma^{k-j} r(s_k^i, a_k^i). \tag{5.2}$$

Then, we simply replace the target with a maximum over the original target and the Monte Carlo target defined in Equation 5.2, given by

$$Q_{\text{MCAC}}^{\text{target}}(s_j^i, a_j^i) = \max \left[ Q^{\text{target}}(s_j^i, a_j^i), Q_{\text{MC-}\infty}^{\text{target}}(s_j^i, a_j^i) \right]. \tag{5.3}$$

This results in the following loss function for training $Q_\theta$:

$$J(\theta) = \ell \left( Q_\theta(s_j^i, a_j^i), Q_{\text{MCAC}}^{\text{target}}(s_j^i, a_j^i) \right). \tag{5.4}$$

The full MCAC training procedure (Algorithm 1) alternates between updating $Q_\theta^\pi$ using the method described above, followed by optimizing the policy $\pi_\phi$ using any standard policy update method.

## 6 Experiments

In the following experiments we study (1) whether MCAC enables more efficient learning when built on top of standard actor-critic RL algorithms and (2) whether MCAC can be applied to improve prior algorithms for RL from demonstrations. See the supplementary material for code and instructions on how to run experiments for reproducing all results in the paper and additional experiments studying the impact of demonstration quantity, quality, and other algorithmic choices such as whether to pretrain learned networks on demonstration data before online interaction.

### 6.1 Experimental Procedure

All experiments were run on a set of 24 Tesla V100 GPUs through a combination of Google Cloud resources and a dedicated lab server. We aggregate statistics over 10 random seeds for all experiments, reporting the mean and standard error across the seeds with exponential smoothing. Details on hyperparameters and implementation details are provided in the supplementary material.

### 6.2 Domains

We consider the five long-horizon continuous control tasks shown in Figure 1. All tasks have relatively sparse rewards, making demonstrations critical for performance. We found that without demonstrations, SAC and TD3 made little to no progress on these tasks.

**Pointmass Navigation:** The first domain is a pointmass 2D navigation task (Figure 1(a)) with time horizon $T = 100$, where the objective is to navigate around the red barrier from start set $\mathcal{S}$ to a goal set $\mathcal{G}$ by executing 2D delta-position controls. If the agent collides with the barrier it receives a reward of $-100$ and the episode terminates. At each time step, the agent receives a reward of $-1$ if it is not in the goal set and $0$ if it is in the goal set. To increase the difficulty of the task, we perturb the state with zero-mean Gaussian noise at each timestep. The combination of noisy transitions and sparse reward signal makes this a very difficult exploration task where the agent must learn to make it through the slit without converging to the local optima of avoiding both the barrier and the slit.

The demonstrator is implemented as a series of proportional controllers which guide it from the starting set to the slit, through the slit, and to the goal set. The actions are clipped to fit in the action space, and trajectories are nearly optimal. The agent is provided with 20 demonstrations.

**Object Manipulation in MuJoCo:** We next consider two object manipulation tasks designed in the MuJoCo physics simulator [Todorov et al., 2012], where the objective is to extract a block from a tight configuration on a table (Block Extraction, Figure 1(b)) and push each of 3 blocks forward on the plane (Sequential Pushing, Figure 1(c)). In the Block Extraction task, the action space consists of 3D delta position controls and an extra action dimension to control the degree to which the gripper is opened. In the Sequential Pushing environment, this extra action dimension is omitted and the gripper is kept closed. In the Block Extraction domain, the agent receives a reward of $-1$ for every timestep that it hasn't retrieved the red block and $0$ when it has. In the Sequential Pushing domain, the reward increases by 1 for each block the agent pushes forward. Thus, the agent receives a reward of $-3$ when it has made no progress and $0$ when it has completed the task. The Block Extraction task is adapted from Thananjeyan et al. [2021] while the Sequential Pushing task is adapted from Wilcox et al. [2021]. We use a time horizon of $T = 50$ for the Block Extraction task and a longer $T = 150$ for the Sequential Pushing task since it is of greater complexity.

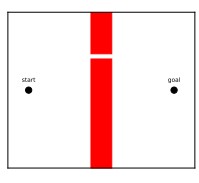 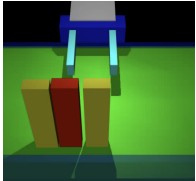 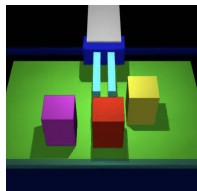 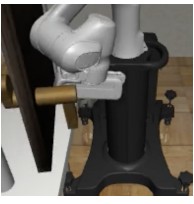 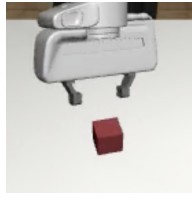

(a) Pointmass Navigation    (b) Block Extraction    (c) Sequential Pushing    (d) Door Opening    (e) Block Lifting

Figure 1: MCAC Domains: We evaluate MCAC on five continuous control domains: a pointmass navigation environment, and four high-dimensional robotic control domains. All domains are associated with relatively unshaped reward functions, which only indicate constraint violation, task completion, or completion of a subtask.

The block extraction demonstrator is implemented as a series of proportional controllers guiding the arm to a position to grip the block, followed by an instruction to close the gripper and a controller to lift. We provide the agent with 50 demonstrations with zero-mean noise injected in the controls to induce suboptimality. For the sequential pushing environment, the demonstrator uses a series of proportional controllers to slowly push one block forward, move backwards, line up with the next block, and repeat the motion until all blocks have been pushed. Because it moves slowly and moves far back from each block it pushes, demonstrations are very suboptimal. For Sequential Pushing, the agent is provided with 500 demonstrations due to the increased difficulty of the task.

**Robosuite Object Manipulation:** Finally, we consider two object manipulation tasks built on top of Robosuite [Zhu et al., 2020], a collection of robot simulation tasks using the MuJoCo physics engine. We consider the Door Opening task (Figure 1(d)) and the Block Lifting task (Figure 1(e)). In the Door Opening task, a Panda robot with 7 DoF and a parallel-jaw gripper must turn the handle of a door in order to open it. The door's location is randomized at the start of each episode. The agent receives a reward of -1 if it has not opened the door and a reward of 0 if it has. In the Block Lifting task, the same Panda robot is placed in front of a table with a single block on its surface. The robot must pick up the block and lift it above a certain threshold height. The block's location is randomized for each episode and the agent receives a reward of $-1$ for every timestep it has not lifted the block and a reward of 0 when it has. Both Robosuite tasks use a time horizon of $T = 50$.

For both Robosuite tasks, demonstrators are trained using SAC on a version of the task with a hand-designed dense reward function, as in the Robosuite benchmarking experiments [Zhu et al., 2020]. In order to ensure suboptimality, we stop training the demonstrator policy before convergence. For each Robosuite environment we use the trained demonstrator policies to generate 100 suboptimal demonstrations for training MCAC and the baselines.

### 6.3 Algorithm Comparisons

We empirically evaluate the following baselines both individually and in combination with MCAC. All methods are provided with the same demonstrations which are collected as described in Section 6.2. See the supplement for more in depth details on implementation and training.

**Behavior Cloning:** Direct supervised learning on the offline suboptimal demonstrations.

**Twin Delayed Deep Deterministic Policy Gradients (TD3) [Fujimoto et al., 2018]:** State of the art actor-critic algorithm which trains a deterministic policy to maximize a learned critic.

**Soft Actor-Critic (SAC) [Haarnoja et al., 2018]:** State of the art actor-critic algorithm which trains a stochastic policy which maximizes a combination of the $Q$ value of the policy and the expected entropy of the policy to encourage exploration.

**Generalized $Q$ Estimation (GQE):** A complex return estimation method from Schulman et al. [2016] for actor-critic methods, which computes a weighted average over TD-$i$ estimates for a range of $i$. GQE is implemented on top of SAC (see supplement for more details). We tune the range of horizons considered and the eligibility trace value (corresponding to $\lambda$ in GAE).

**Overcoming Exploration from Demonstrations (OEFD) [Nair et al., 2018]:** OEFD builds on DDPG [Lillicrap et al., 2015] by adding a loss which encourages imitating demonstrations and a

learned filter which determines when to activate this loss. Our implementation does not include hindsight experience replay since it is not applicable to most of our environments.

**Conservative $Q$ Learning (CQL) [Kumar et al., 2020]:** A recent offline RL algorithm that addresses value overestimation with a conservative $Q$ function. Here we also update CQL online after pre-training offline on demonstrations in order to provide a fair comparison with other algorithms.

**Advantage Weighted Actor-Critic (AWAC) [Nair et al., 2021]:** A recent offline reinforcement learning algorithm designed for fast online fine-tuning.

We also implement versions of each of the above RL algorithms with MCAC (TD3 + MCAC, SAC + MCAC, GQE + MCAC, OEFD + MCAC, CQL + MCAC, AWAC + MCAC).

The behavior cloning comparison serves to determine whether online learning is beneficial in general, while the other comparisons study whether MCAC can be used to accelerate reinforcement learning for commonly used actor-critic algorithms (TD3, SAC, and GQE, which is essentially SAC with complex returns) and for recent algorithms for RL from demonstrations (OEFD, CQL and AWAC).

## 6.4 Results

In Section 6.4.1, we study MCAC on a simple didactic environment to better understand how it affects the learned Q-values. We then study whether MCAC can be used to accelerate exploration on a number of continuous control domains. In Section 6.4.2, we study whether MCAC enables more efficient learning when built on top of widely used actor-critic RL algorithms (SAC, TD3, and GQE). Then in Section 6.4.3, we study whether MCAC can provide similar benefits when applied to recent RL from demonstration algorithms (OEFD, CQL, and AWAC). Additionally, in the supplement we provide experiments involving other baselines, investigating the sensitivity of MCAC to the quality and quantity of demonstration data, and investigating its sensitivity to pretraining.

### 6.4.1 MCAC Didactic Example

In order to better understand the way MCAC affects $Q$ estimates, we visualize $Q$ estimates when MCAC is applied to SAC after 50000 timesteps of training in the Pointmass Navigation environment, in Figure 2. Here we visualize $Q$-values for the entire replay buffer, including offline demonstrations,

When training without MCAC (top row), the agent is unable to learn a useful $Q$ function and thus does not learn to complete the task (the only successful trajectories shown are offline data). However, even when this is the case, the MCAC estimate is able to effectively propagate reward signal backwards along the demonstrator trajectories, predicting higher rewards early on (top right). We see that the GQE estimates (top middle) are somewhat more effective than the Bellman ones at propagating

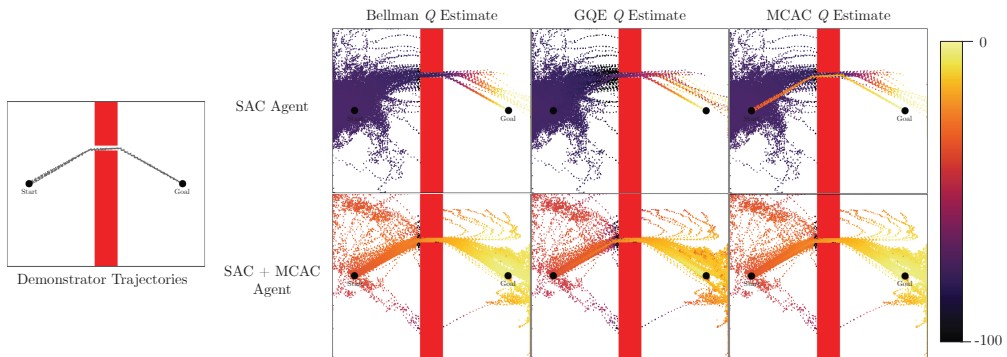

Figure 2: **MCAC Replay Buffer Visualization:** Scatter plots showing Bellman, GQE and MCAC $Q$ estimates on the entire replay buffer, including offline demonstrations, for SAC learners with and without the MCAC modification after 50000 timesteps of training. The top row shows data and $Q$ estimates obtained while training a baseline SAC agent without MCAC, while the bottom row shows the same when SAC is trained with MCAC. The left column shows Bellman $Q$ estimates on each replay buffer sample while the middle column shows GQE estimates and the right column shows MCAC estimates. Results suggest that MCAC is helpful for propagating rewards along demonstrator trajectories.

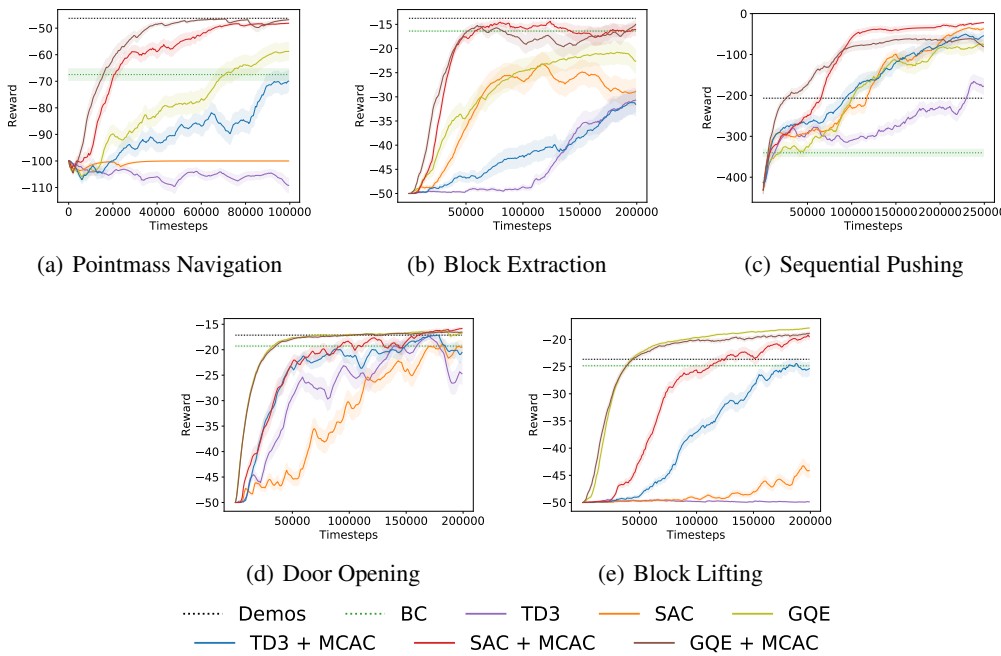

(a) Pointmass Navigation     (b) Block Extraction     (c) Sequential Pushing

(d) Door Opening     (e) Block Lifting

Figure 3: **MCAC and Standard RL Algorithms Results:** Learning curves showing the exponentially smoothed (smoothing factor $\gamma = 0.9$) mean and standard error across 10 random seeds. We find that MCAC improves the learning efficiency of TD3, SAC, and GQE across all 5 environments.

reward, but not as effective as MCAC. When the agent is trained with MCAC (bottom row), the agent learns a useful $Q$ function that it uses to reliably complete the task (bottom left). As expected with a high-performing policy, its Bellman estimates, GQE estimates and MCAC estimates are similar.

### 6.4.2    MCAC and Standard RL Algorithms

In Figure 3, we study the impact of augmenting SAC, TD3 and GQE with the MCAC target $Q$-function. Note that all methods, both with and without MCAC, we initialize their replay buffers with the same set of demonstrations. Results suggest that MCAC is able to accelerate learning for both TD3 and SAC across all environments, and is able to converge to performance either on-par with or better than the demonstrations. In the Pointmass Navigation and Block Lifting tasks, SAC and TD3 make no task progress without MCAC. MCAC also accelerates learning for GQE for the Pointmass Navigation, Block Extraction, and Sequential Pushing environments. In the Door Opening and Block Lifting environments, MCAC leaves performance largely unchanged since GQE already achieves performance on par with the next best algorithm without MCAC.

### 6.4.3    MCAC and RL From Demonstrations Algorithms

In Figure 4, we study the impact of augmenting OEFD [Nair et al., 2018], CQL [Kumar et al., 2020] and AWAC [Nair et al., 2021] with the MCAC target $Q$-function. Results suggest that MCAC improves the learning efficiency of OEFD on the Pointmass Navigation, Sequential Pushing, and Block Lifting tasks, but does not have a significant positive or negative affect on performance for the Block Extraction and Door Opening tasks. MCAC improves the performance of AWAC on the Pointmass Navigation and Sequential Pushing environments, stabilizing learning while the versions without MCAC see performance fall off during online fine tuning. On the other 3 environments where AWAC is able to immediately converge to a stable policy after offline pre-training, MCAC has no significant negative effect on its performance. In all tasks, MCAC improves the performance of CQL. In particular, for the Pointmass Navigation, Block Extraction and Sequential Pushing tasks, CQL makes almost no progress while the version with MCAC learns to complete the task reliably.

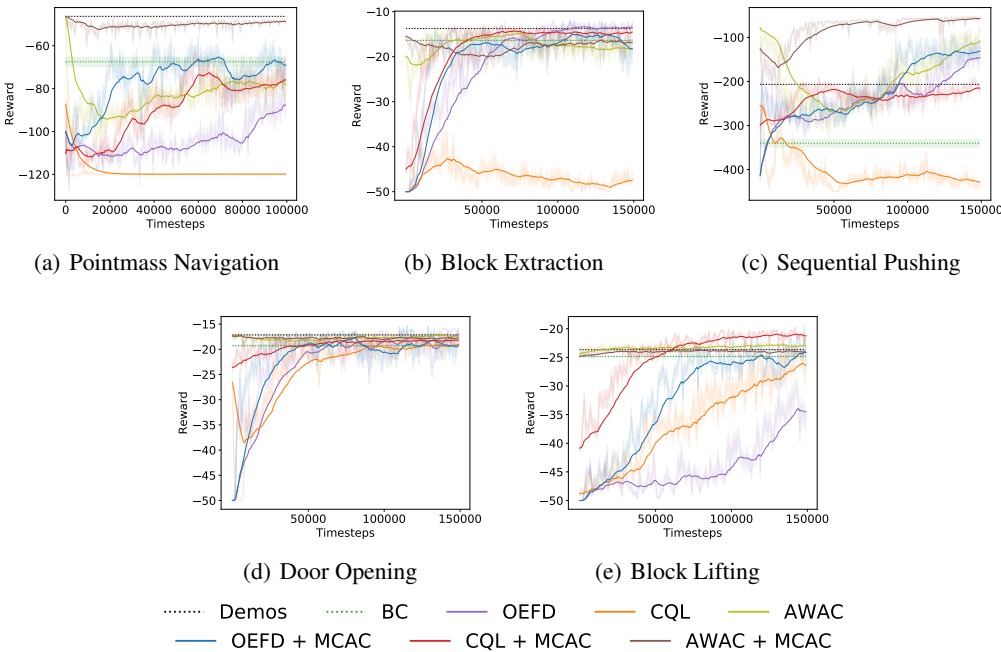

| | | | | |
|---|---|---|---|---|
| (a) Pointmass Navigation | | (b) Block Extraction | | (c) Sequential Pushing |

| | | |
|---|---|---|
| (d) Door Opening | | (e) Block Lifting |

Demos ........ BC ........ OEFD ─── CQL ─── AWAC ─── OEFD + MCAC ─── CQL + MCAC ─── AWAC + MCAC

Figure 4: **MCAC and RL from Demonstrations Algorithm Results:** Learning curves showing the exponentially smoothed (smoothing factor $\gamma = 0.9$) mean and standard error across 10 random seeds. When OEFD or AWAC achieve high performance almost immediately, MCAC has little impact on performance. However, when OEFD and AWAC are unable to learn efficiently, MCAC accelerates and stabilizes policy learning.

## 7 Discussion, Limitations, and Ethical Considerations

We present Monte Carlo augmented Actor-Critic (MCAC), a simple, yet highly effective, change that can be applied to any actor-critic algorithm in order to accelerate reinforcement learning from demonstrations for sparse reward tasks. We present empirical results suggesting that MCAC often significantly improves performance when applied to three state-of-the-art actor-critic RL algorithms and three RL from demonstrations algorithms on five different continuous control domains.

Despite strong empirical performance, MCAC also has some limitations. We found that while encouraging higher $Q$-value estimates is beneficial for sparse reward tasks, when rewards are dense and richly informative, MCAC is not helpful and can even hinder learning by overestimating $Q$-values. From an ethical standpoint, reinforcement learning algorithms such as MCAC can be used to automate aspects of a variety of interactive systems, such as robots, online retail systems, or social media platforms. However, poor performance when policies haven't yet converged, or when hand-designed reward functions do not align with true human intentions, can cause significant harm to human users. Lastly, while MCAC shows convincing empirical performance, in future work it would be interesting to provide theoretical analysis on its convergence properties and on how MCAC can be used to learn $Q$ functions which better navigate the bias-variance tradeoff.

## Acknowledgements

We would like to thank all reviewers for their invaluable feedback, which helped us significantly strengthen the paper during the author response period. This research was performed at the AUTOLAB at UC Berkeley in affiliation with the Berkeley AI Research (BAIR) Lab, and the CITRIS "People and Robots" (CPAR) Initiative. The authors were supported in part by donations from Google, Toyota Research Institute, and NVIDIA.

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
