# OpenReview forum: "Monte Carlo Augmented Actor-Critic for Sparse Reward Deep Reinforcement Learning from Suboptimal Demonstrations"
_NeurIPS.cc/2022/Conference — NeurIPS 2022 Accept_

### Official Review · Reviewer_rJu7 · 2022-07-10

**Rating:** 5
**Confidence:** 4
**Soundness:** 2 fair
**Presentation:** 3 good
**Contribution:** 3 good

**Summary:**

This paper proposes a modified Actor-Critic algorithm (MCAC) for sparse reward problems. This modification is achieved by initializing the replay buffer with demonstration and computing the modified Q-value by combining temporal distance target and Monte Carlo estimation. The method is evaluated in five environments, and performance is compared with a few existing methods.

**Questions:**

Standard RL algorithms usage demonstration (line 297) and is then compared with the MCAC version. However, the standard RL is not equipt to deal with such a demonstration, and having some sub-optimal demonstration might worsen the performance. Thus, RL algorithms without the sub-optimal demonstration should be compared as well.

In figure 3, MCAC barely improves performance in all the environments; in most scenarios, performance remains the same as the starting position (initial timesteps); why is that the case, especially AWAC and AWAC+MCAC?

Additionally, the results seem puzzling ( Figure 3); the AWAC results start with a high reward and then fall. In the cases where it recovers (with MCAC) and barely reaches the initial performance again. Thus though MCAC makes training stable, it barely gives any reward-level advantage. This improvement is the core of this paper. Though on OEFD, MCAC is performing better; overall, this method appears to be not as good as AWAC-based methods.

How was the demonstration collected? Was it collected by programming (automatic) or human-collected (e.g., line 239)? Adding zero-mean noise might make it a bit realistic; however, it may be beneficial to have humans collect data, which would be more realistic.

I understand that the paper focus on suboptimal demonstration (e.g., line 258); however, is the proposed method tied to such an assumption? How does the performance stand with near-to optimal demonstration, which is often the case of expert (human) demonstration?

Looking at Figures 2 and 3, it seems the demonstration-based method performs worse than the standard-baseline variant (e.g., GQE in Figure 2). It seems that the demonstration is not helping in these scenarios. Further discussion on this issue would help clarify the advantage of the proposed method.

The MCAC has two modifications, namely, replay buffer and Q-value modification. An ablation study on them would further clarify their significance.

There are many missing references (“?”) throughout the paper. A careful pass should fix them. There are too many environmental descriptions in the main paper, which can go to the Appendix.


**Limitations:**

Yes. Limitations and potential societal impacts are discussed.

**Strengths And Weaknesses:**

### Strength:
- Tackle an important problem of learning from sub-optimal demonstration in sparse environments.
- Relevant literature is discussed, and a few baselines are compared in five environments.

### Weakness (Details in Questions):
- Lack of empirical evidence of the advantage of the proposed method
- Some details of the method design and experiments are missing.

---

> ### Author Response · Authors · 2022-08-02
> **Response to Reviewer rJu7**
>
> We thank reviewer rJu7 for the time spent reviewing and their valuable feedback. We have uploaded a new version of the manuscript and supplementary material in which we've addressed reviewer concerns, with text that has been added since the original submission updated to be blue. This response has been split into multiple parts due to space constraints.
>
> >Standard RL algorithms usage demonstration (line 297) and is then compared with the MCAC version. However, the standard RL is not equipt to deal with such a demonstration, and having some sub-optimal demonstration might worsen the performance. Thus, RL algorithms without the sub-optimal demonstration should be compared as well.
>
> The only difference between MCAC and the standard RL algorithms we consider is the change to the Q-function update in MCAC. For example, SAC and SAC + MCAC both are seeded with demonstrations in their replay buffer, with the only difference between that SAC + MCAC performs the SAC update by taking a maximum over the standard Bellman targets and Monte Carlo targets. As a result, we argue that both algorithms have equal mechanisms to utilize demonstrations to accelerate learning, with the only difference between the two being the MCAC update.
>
> For the tasks we consider, since the rewards are exceedingly sparse, we found that standard RL algorithms are usually unable to make much progress at all without demonstrations added to the replay buffer, which is why we initially omitted this comparison. However, we agree that this comparison is still important to confirm that demonstrations are indeed needed to make progress in the tasks we consider. To this end, we added experiments in section B.1 of the supplement which omit demonstration data. For most environments, we confirm that the demonstrations were necessary for the learner to make any progress as the versions without demonstrations fail to complete the task ever. The one exception is the sequential pushing environment, where the agents without demonstrations still learn to make some progress, although much more slowly than with demonstrations. Based on the fact that the variants without demonstration data perform substantially worse overall than the ones in the paper, we are confident that the versions in the paper are the appropriate baselines.
>
> >In figure 3, MCAC barely improves performance in all the environments; in most scenarios, performance remains the same as the starting position (initial timesteps); why is that the case, especially AWAC and AWAC+MCAC? Additionally, the results seem puzzling ( Figure 3); the AWAC results start with a high reward and then fall. In the cases where it recovers (with MCAC) and barely reaches the initial performance again. Thus though MCAC makes training stable, it barely gives any reward-level advantage. This improvement is the core of this paper. Though on OEFD, MCAC is performing better; overall, this method appears to be not as good as AWAC-based methods.
>
> In these cases (figures 2d and 2e) the baselines already learn very good policies, and there is not much room for improvement. However, notice that in figures 2a-c, MCAC is able to stabilize performance while AWAC falls off. We believe this fall off is likely due to the sudden change in distribution between the offline data AWAC is initially trained on and the online data added to the replay buffer during online interaction. We find that MCAC however is able to stabilize learning significantly by ensuring that the Q-values of transitions in online data do not drop too much. We do not claim that adding MCAC will make an algorithm strictly better, but empirical results suggest that it is always at least as good as the original baseline, and often is much better.
>
> Looking at the Q values for the runs of AWAC with and without MCAC, we noticed that without MCAC the Q values quickly plummeted during online training after unsuccessful trajectories were added to the replay buffer, whereas the version with MCAC did not have this issue. Although further work is necessary to fully confirm this, we believe that this phenomenon plays a large part in the difference. We added this analysis to the supplement in Section B.7. As a side note, we believe this experiment opens doors for exciting new research on the effect of the MCAC objective on distribution shift in RL, an area currently attracting much interest in the ML research community. In future work, we’ll further investigate this phenomenon to see whether MCAC is helpful for this problem.

---

> > ### Author Response · Authors · 2022-08-02
> > **Response to Reviewer rJu7 (part 2)**
> >
> > >How was the demonstration collected? Was it collected by programming (automatic) or human-collected (e.g., line 239)? Adding zero-mean noise might make it a bit realistic; however, it may be beneficial to have humans collect data, which would be more realistic.
> >
> > We describe how demonstrations are collected for all environments in Section 6.2. The demonstrations were collected in a programmatic manner as is consistent with prior work in RL from demonstrations such as the papers which proposed OEFD (Nair et al., 2018) and AWAC (Nair et al. 2021). We agree that working with real human demonstrations would also be interesting though and hope to explore this in future work.
> >
> > >I understand that the paper focuses on suboptimal demonstration (e.g., line 258); however, is the proposed method tied to such an assumption? How does the performance stand with near-to optimal demonstration, which is often the case of expert (human) demonstration?
> >
> > In figure 2a in the supplementary material we study the effect that demonstration quality has on learner performance. Overall, the agent performs better when demonstrations are closer to optimality. However, we emphasize that since the algorithm doesn't strictly tie performance to the demonstrations, it is able to surpass demonstrator performance (as described at the end of section 5.1), in some environments learning near optimal policies despite suboptimal demonstrations (figure 2c,e, 3c).
> >
> > >Looking at Figures 2 and 3, it seems the demonstration-based method performs worse than the standard-baseline variant (e.g., GQE in Figure 2). It seems that the demonstration is not helping in these scenarios. Further discussion on this issue would help clarify the advantage of the proposed method.
> >
> > While GQE is not explicitly an RL from demonstrations algorithm, prior work (Vecerik et al., 2018) makes clear that n-step methods such as GQE are helpful for learning from demonstrations. Thus, it makes sense that in some cases it might perform better than the demonstration-focused algorithms in figure 3.
> >
> > We would additionally clarify that since all algorithms start with a replay buffer seeded with demonstrations, the demonstrations are helping in all scenarios, as shown in Figure 1 in the supplement.
> >
> > >The MCAC has two modifications, namely, replay buffer and Q-value modification. An ablation study on them would further clarify their significance.
> >
> > We already have ablations regarding the Q-value modification since all of the RL algorithms on their own (eg. SAC, TD3, GQE, OEFD, AWAC) do not have the Q-value modification that MCAC implements, but still include demonstrations in the replay buffer. As mentioned earlier, we did not initially include results without any demonstrations in the replay buffer because the tasks we consider have very sparse rewards, making it hard for standard RL algorithms to make much progress. However, we agree that this comparison is still important to confirm that demonstrations are indeed needed to make progress in the tasks we consider. To this end, we added experiments to Section B.1 in the supplement evaluating the learners without demonstrations, where we see that in all but one of the environments the learners fail to make any progress. In the sequential pushing environment the learner without demonstrations does make some progress, although much more slowly than the variant with demonstrations. In this experiment we see that the learner performs roughly the same with and without MCAC, but in future work we’ll study more environments where it is possible to make progress without demonstrations to see whether MCAC still has a benefit. In summary, our ablations provide empirical evidence that both modifications (replay buffer and Q-value modification) are critical to the success of MCAC.
> >
> > >There are many missing references (“?”) throughout the paper. A careful pass should fix them.
> >
> > We apologize for this oversight, all of these missing references have now been fixed.

---

> > > ### Comment · Reviewer_rJu7 · 2022-08-08
> > > **Response**
> > >
> > > I appreciate the authors for the detailed response. With the updated version, the experimental details are improved compared to the initial version of the paper. However, my concern regarding the lack of empirical evidence of the advantage of the proposed method still remains. Therefore, I increase my score from 4 to 5.

---

> > > > ### Author Response · Authors · 2022-08-08
> > > > **Thank you for updating your score! We would appreciate details on remaining concerns.**
> > > >
> > > > Thank you for considering our response and updating your score accordingly. If you might be able to provide details regarding any remaining concerns you have regarding "the lack of empirical evidence of the advantage of the proposed method", we would be happy to conduct further experiments or provide additional clarifications to help address these points.

---

### Official Review · Reviewer_dRfP · 2022-07-10

**Rating:** 5
**Confidence:** 4
**Soundness:** 3 good
**Presentation:** 2 fair
**Contribution:** 2 fair

**Summary:**

This paper proposes a simple yet effective method MCAC to assimilate the benefits of both TD Q-target and Monte Carlo Q-targets, which can be applied to any actor-critic algorithm to solve the sparse Reinforcement learning problem. The authors empirically prove the effectiveness of MCAC for both standard reinforcement learning and imitation learning.

**Questions:**

For the experimental parts, the authors claim in Figure 2 that when GQE already achieves very strong performance, MCAC may have little impact. I feel confused about the criterion of "strong performance". If strong performance refers to the performance being higher than the demos, we can still see that on the Sequential Pushing, although GQE achieves strong performance, MCAC can still accelerate the learning.

I also have a similar concern for Figure 3 about OEFD, the criterion for "unable to learn efficiently or have significant instability".

Another stuff is for Figure 3 in the appendix, I cannot see a clear pattern as $\lambda$ increases (e.g., block lifting), I feel the results are still dominated by the randomness rather than the hyperparameters.

My general intuition is that MCAC is not shown to consistently take effect in these 5 environments, I suggest the authors give some theoretical proof and more experimental results on other domains.

**Limitations:**

The authors address both societal impacts and limitations.

**Strengths And Weaknesses:**

Pros:
The method proposed is simple and works effectively on various tasks. For some tasks, MCAC achieves a pretty solid performance improvement whereas the baselines work very badly. The authors also provide some intuitive explanations for MCAC which I think is inspiring for future work.

Cons:
However, my largest concern is that either the current experimental results or the intuitive explanations cannot justify the consistent effectiveness of MCAC, and generally, I think this work does not show enough technical contribution. This method is just a simple extension the idea in [1]. Besides, the writing of this paper is very messy with many missing section references. I see the authors want to give some ablation study in section 6.4 but that part is finally in the appendix.

[1] Robert Wright, Steven Loscalzo, Philip Dexter, and Lei Yu. Exploiting multi-step sample trajectories
419 for approximate value iteration. In Hendrik Blockeel, Kristian Kersting, Siegfried Nijssen, and
420 Filip Železný, editors, Machine Learning and Knowledge Discovery in Databases, pages 113–128,
421 Berlin, Heidelberg, 2013. Springer Berlin Heidelberg. ISBN 978-3-642-40988-2.

---

> ### Author Response · Authors · 2022-08-02
> **Response to Reviewer dRfP**
>
> We thank reviewer dRfP for the time spent reviewing and their valuable feedback. We have uploaded a new version of the manuscript and supplementary material in which we've addressed reviewer concerns, with text that has been added since the original submission updated to be blue. This response has been split into multiple parts due to space constraints.
>
> >The current experimental results or the intuitive explanations cannot justify the consistent effectiveness of MCAC, and generally, I think this work does not show enough technical contribution.
>
> Thanks for the concern. As far as we can tell we’ve provided clear empirical evidence that MCAC is very effective in the problem settings we consider, but if you believe there are ways that we can improve upon this we would love to use your feedback to improve our paper. To this end, we have provided evidence that MCAC helps boost performance of two widely used actor critic algorithms (SAC and TD3), a complex return variant of SAC (GQE), and two SOTA RL from demonstrations algorithms (OEFD and AWAC)  across 5 different continuous control tasks. If you could please be more specific with your reasons that you don’t believe our claims are sound that would be very helpful.
>
> >This method is just a simple extension the idea in [1].
>
> We would like to point out that [1] only considers learning with low-dimensional function approximators and addresses fairly simple environments. Additionally, the method is only shown to work in an offline fashion. In this work, we show how to scale the idea in the paper for learning to complete difficult continuous control robotic manipulation tasks given only a sparse reward signal by using demonstrations in an online fashion.
>
> Additionally, we would point out that due to the scope considered and the paper's age, the method no longer receives much attention, despite the fact that, as we’ve shown in this paper, it is still useful. We hope that by publishing this paper the idea can gain more attention in the deep RL research community.
>
> Finally, while we do concede that our method is “simple”, we would argue that simple ideas, when they work well, can be very impactful. We believe our empirical results make a strong case that our method is useful.
>
> >The writing of this paper is very messy with many missing section references. I see the authors want to give some ablation study in section 6.4 but that part is finally in the appendix.
>
> Apologies for the reference issues, and thank you for pointing them out. They've been addressed in the updated version of the manuscript. We are not sure what else in the writing is messy, and would appreciate specific clarification on this so that we can fix any additional issues.
>
> >For the experimental parts, the authors claim in Figure 2 that when GQE already achieves very strong performance, MCAC may have little impact. I feel confused about the criterion of "strong performance". If strong performance refers to the performance being higher than the demos, we can still see that on the Sequential Pushing, although GQE achieves strong performance, MCAC can still accelerate the learning. I also have a similar concern for Figure 3 about OEFD, the criterion for "unable to learn efficiently or have significant instability".
>
> The key takeaway from the results is that MCAC, when added to existing actor critic algorithms, almost always provides a boost in performance and seems to never significantly hurt performance in the domains we considered. We show this across 5 different algorithms (SAC, TD3, GQE, OEFD, AWAC) and across 5 different tasks. Thus, while it is true that in settings where GQE already does well, MCAC does not provide much additional benefit (because there is not much room to improve), in settings where GQE does poorly such as the navigation, block extraction, and pushing environments, MCAC provides a significant performance boost (see Figures 2a-c). A similar pattern can be found in Figure 3 for OEFD, where MCAC generally helps performance when OEFD performs poorly and does not harm performance when OEFD already is able to learn quickly. We agree that the terms “strong performance” and “significant instability” are a bit vague in this context and have updated the writing in Sections 6.4.1 and 6.4.2 to rectify this. We apologize for this confusion.
>
> >Another stuff is for Figure 3 in the appendix, I cannot see a clear pattern as λ increases (e.g., block lifting), I feel the results are still dominated by the randomness rather than the hyperparameters.
>
> The λ experiment was meant to serve as an additional baseline and we did not include it in the main paper because of the unexplainable effects of λ, opting instead to include GQE, another n-step method that was more principled.

---

> > ### Author Response · Authors · 2022-08-02
> > **Response to Reviewer dRfP (part 2)**
> >
> > >My general intuition is that MCAC is not shown to consistently take effect in these 5 environments, I suggest the authors give some theoretical proof and more experimental results on other domains.
> >
> > The method's benefit relies on deep function approximation, which is notoriously difficult to perform theoretical analysis on. However, the empirical results show that our method has no clear disadvantages and often has a clear benefit in all the environments we study. We believe our empirical evidence for MCAC’s efficacy is quite thorough, as we include results on 5 environments across 5 algorithms and show that MCAC consistently boosts performance (or leaves it unchanged when performance is already good) across all domains for all algorithms.

---

> > > ### Comment · Reviewer_dRfP · 2022-08-07
> > > **Response to the authors**
> > >
> > > Thank you for your response to my questions.
> > >
> > > After reading the responses and updated version, I think the authors addressed most of my concerns. I agree with the authors that
> > >
> > > > The key takeaway from the results is that MCAC, when added to existing actor critic algorithms, almost always provides a boost in performance and seems to never significantly hurt performance in the domains we considered.
> > >
> > > But I mainly want to know if the authors could give an intuition when MCAC could boost the performance. From my point of view, the authors should try to distinguish the case that "GQE already does well" and "MCAC has no negative effect". How can we tell if GQE has already done well or just poorly? It will be much better if the authors could give a more clear sense of that.
> > >
> > > Basically, I think the current claims in the updated version make sense and I'm willing to raise my score.

---

> > > > ### Author Response · Authors · 2022-08-08
> > > > **Thanks for your response! Further intuition on MCAC is provided below.**
> > > >
> > > > Thank you for your detailed consideration of our response. We agree that given that MCAC provides a simple mechanism to boost performance for a number of actor critic algorithms across a number of tasks, further intuition on when MCAC can boost performance would be very valuable.
> > > >
> > > > The key intuition is that MCAC is particularly valuable for boosting performance in sparse reward environments because the Monte Carlo targets help propagate reward information over long-horizons more effectively early on in learning than TD targets, making it possible to prevent underestimation of Q-values in the neighborhood of successful trajectories. However, studying precisely when MCAC may help boost performance of specific algorithms in sparse reward settings (eg. GQE) is particularly challenging because it requires analyzing precisely how MCAC would change the learning dynamics of a deep RL algorithm. Deep RL algorithms have been notoriously hard to theoretically analyze, leading to little relevant results in prior work. However, we agree that this would be a very exciting direction to study in the future.

---

### Official Review · Reviewer_VGAN · 2022-07-11

**Rating:** 6
**Confidence:** 4
**Soundness:** 3 good
**Presentation:** 3 good
**Contribution:** 3 good

**Summary:**

This paper proposes an Actor-Critic method (MCAC) to resolve the sparse reward problem in deep reinforcement learning (single-agent cases). Technically, the algorithm starts from a replay buffer with demonstrations and performs policy evaluation with a modified $Q$-value function. Specifically, the policy evaluation is the maximum of the standard critic function and a Monte-Carlo estimation. The idea behind using two value estimations is that the combination may share the advantages of these two estimations, such as low variance and unbias. The implementation is given under some assumptions (such as "... the last observed reward value will repeat forever ...") to make the algorithm adapt to cases with an infinite horizon. The authors demonstrate experiments and comparisons with previous work on five continuous control tasks, in which the results show that MCAC outperforms all baselines.

**Questions:**

1. As the MCAC requires pre-training with offline datasets. So, Is there any requirement on the size of the dataset? I think it is critical to the algorithm's performance.
2. For $Q^{target}_{MC-\infty}$ in the infinite cases, why don't we use the critic to approximate the $\gamma^{T-j+1}\frac{r^i_T}{1-\gamma}$?


**Limitations:**

The authors figure out some unresolved limitations in the last section. Such as MCAC is not helpful and even hinders learning by overestimating Q-values in dense-reward cases. As I mentioned in the "weaknesses", the normalization of the maximum operation maybe necessary to handle this problem.

**Strengths And Weaknesses:**

### Update After Rebuttal

I thank the authors for their in-depth responses also the supplemented experiments. I am mostly convinced by the authors and have updated my score from 5 to 6.

### Strengths

1. this paper is well-organized and easy to follow.
2. this paper concentrates on an interesting problem in deep reinforcement learning and proposes a novel plug-in for Actor-Critic methods.

### Weaknesses

1. **Some typos and referring errors**: e.g., line 209: "... provided in Appendix ??"
2. **The demonstration is not so serious**: the demonstration lacks a comparison or investigation of the effectiveness of value function combinations. As the authors claim in this paper, they aim to bridge the advantages of the standard value function and Monte-Carlo methods to improve the accuracy/efficiency of policy evaluation (or value estimation). However, there is no explanation (at least empirically) for why this method can achieve such a target.
3. **Some technique flaws**:

    - As introduced in this paper, MCAC modifies the estimation of the target value function as Eq-(5.3). However, there may be a potential issue that such a method is biased as $Q^{target}$ and $Q^{target}_{MC-\infty}$ have different scales.
    - The authors approximate the Monte-Carlo estimation of an infinite horizon as Eq-(5.2). However, I am concerned about this approximation since the authors neither refer to some previous work to support the correctness nor give empirical results to explain it.
4. This point is more like a suggestion to highlight your motivation. As the authors claim that their method is the first application of Monte-Carlo estimation of the return-to-go to improve value estimation for online cases, while is no discussion about what will such an estimator bring to online cases. Thus, I think the authors can make a deeper discussion on that.

---

> ### Author Response · Authors · 2022-08-02
> **Response to Reviewer VGAN**
>
> We thank reviewer VGAN for the time spent reviewing and their valuable feedback. We have uploaded a new version of the manuscript and supplementary material in which we've addressed reviewer concerns, with text that has been added since the original submission updated to be blue. This response has been split into multiple parts due to space constraints.
>
> >Some typos and referring errors: e.g., line 209: "... provided in Appendix ??"
>
> Apologies for the issues, and thanks for pointing them out. We’ve fixed them in the updated manuscript.
>
> >The demonstration is not so serious: the demonstration lacks a comparison or investigation of the effectiveness of value function combinations. As the authors claim in this paper, they aim to bridge the advantages of the standard value function and Monte-Carlo methods to improve the accuracy/efficiency of policy evaluation (or value estimation). However, there is no explanation (at least empirically) for why this method can achieve such a target.
>
> We are not sure what the reviewer means by “a comparison or investigation of the effectiveness of value function combinations” and would appreciate clarification on this point. We do study GQE, which combines value functions at numerous different time horizons (see Section 6.4.1 and Figure 2) and find that MCAC boosts the performance of this method as well.
>
> >As introduced in this paper, MCAC modifies the estimation of the target value function as Eq-(5.3). However, there may be a potential issue that such a method is biased as Qtarget and QMC−∞target have different scales.
>
> Note that both Qtarget and QMC−∞ target are estimators of the same quantity, but just have different bias/variance properties. Thus, in the limit, they should be on the same scale, and any difference in their values early on is precisely what MCAC is exploiting: namely we expect the QMC−∞ target to be higher initially since it is easier to propagate values from initially successful trajectories but eventually when the agent has enough online experience, the Bellman Qtarget should be better calibrated with the QMC−∞ target. This is confirmed by a study we did on the scales of the different estimators, presented in Section B.6 in the supplement. For successful trajectories we see that early on MCAC estimates are higher than the bellman estimates, which helps to increase Q values along demonstrator trajectories and help the agent learn to find the goal. Then, we see that in the limit the bellman estimates, monte carlo estimates, and MCAC estimates all converge to similar values. We see that for failed trajectories the Monte Carlo estimate always underestimates Q values, but this is canceled out when the MCAC estimate takes the maximum.
>
> >The authors approximate the Monte-Carlo estimation of an infinite horizon as Eq-(5.2). However, I am concerned about this approximation since the authors neither refer to some previous work to support the correctness nor give empirical results to explain it.
>
> We would like to emphasize that in the settings we consider (sparse reward), this estimation makes clear intuitive sense: we assume that if the agent ended up in the goal set it will stay there forever and if it didn't make it there it will never get there. The main drawback of this approach is that it has the potential for underestimation when the agent hits the time horizon just before succeeding, but in these cases the underestimated Q target will be canceled out by MCAC when taking the max against the bellman Q target. This intuition is backed up by the experimental results we added in Section B.6 in the supplement.
>
> >As the authors claim that their method is the first application of Monte-Carlo estimation of the return-to-go to improve value estimation for online cases, while there is no discussion about what will such an estimator bring to online cases. Thus, I think the authors can make a deeper discussion on that.
>
> Thank you for the suggestion. We’ve added some lines to Section 2.2 with a deeper explanation of the motivations for extending the ideas in Wright et al (2013) to the online deep RL setting. First, deep RL algorithms are often unstable, and as we empirically verify in the paper, this method is a promising way to improve this. Second, while offline RL does have important applications, online RL is much more widely studied, so it’s worth understanding how the ideas perform here.
>
> >As the MCAC requires pre-training with offline datasets. So, Is there any requirement on the size of the dataset? I think it is critical to the algorithm's performance.
>
> Thanks for the question! We did a study on this, and the results are presented in Figure 2b in the supplementary material. We have also added a reference to these experiments in Section 6.4 for further clarity.

---

> > ### Author Response · Authors · 2022-08-02
> > **Response to Reviewer VGAN (part 2)**
> >
> > >For QMC−∞ target in the infinite cases, why don't we use the critic to approximate the γT−j+1rTi1−γ?
> >
> > Thank you for this interesting idea! We ran this experiment and added the results to Section B.5 in the supplement. We found that in most experiments this resulted in roughly the same performance. However, there were numerous settings where the original estimator outperformed the version using the critic. Although further work is required to fully understand why this might be the case, we note that in successful trajectories the original estimator is optimistic while for failed ones it is pessimistic, which combines well with the MCAC objective to direct exploration towards successful trajectories without adding too much variance to failed ones.

---

> > > ### Comment · Reviewer_VGAN · 2022-08-06
> > > **Thanks for your responses**
> > >
> > > I thank the authors for their in-depth responses, also the supplemented experiments. They resolved most of my questions and improved the clarity of their work. As for the "comparison of the effectiveness of value function combinations," I mean that the authors should conduct some numeric experiments or theoretical analysis to support their claim to "reduce the bias and variance properties of the Monte Carlo and temporal difference Q-targets". Still, I don't think that should be a reason to reject the paper, and could be a topic for future work.

---

> > > > ### Author Response · Authors · 2022-08-06
> > > > **Thank you for updating your score!**
> > > >
> > > > Thank you for considering our response and updating your score! We found that studying MCAC theoretically from a bias-variance perspective was quite challenging when combined with deep function approximation, and have added a brief discussion of this point to the limitations section of the paper. We agree that this is an exciting area for us to study in future work.

---

### Official Review · Reviewer_jXtd · 2022-07-12

**Rating:** 6
**Confidence:** 4
**Soundness:** 3 good
**Presentation:** 3 good
**Contribution:** 3 good

**Summary:**

The paper proposes a method to incorporate prior expert (possibly sub-optimal) demonstrations in the learning procedure of any actor-critic algorithms. Furthermore, the authors propose also a modified target estimator for the critic updates, based on the maximum of an MC estimate and the standard estimators of the base AC algorithms. The resulting framework is fairly easy to implement and introduces no additional parametrization or computation at the cost of an initial set of demonstrations from the expert. The empirical evaluation show an improvement over SOTA RL algorithms on a set of continuous domains with sparse rewards.

**Questions:**

The main question I have is how important is the presence of the expert demonstrations in the replay buffer. Namely, have you measured the performance of the proposed algorithm only with the application of the MC targets in Eq. 5.4?

Related to this, a study on the effect of the sub-optimality of the demonstrations should be performed. How much is the performance impacted from more sub-optimal demonstrations. This could be studied in a simple environment, by varying the amount of noise added to the expert demonstrations. This is important, since the method, as the authors state, is to be applied exclusively to sparse reward environments, and often times in these environments it might be difficult to have demonstrations with a reward signal (e.g. goal based environments where a human might have difficulties in making a policy that reaches the goal)

**Limitations:**

The authors adequately address their limitations in the conclusions.

**Strengths And Weaknesses:**

The paper is clearly written and easy to understand. The authors propose a novel method to for Q-target correction, which is fearly easy to combine with current RL algorithms. The experimental evaluation shows significant performance improvement without additional computational cost.

Overall I like the paper and I think it should be accepted.  The authors propose a simple method, thoroughly evaluate it and compare with appropriate baselines and discuss the limitations and shortcomings of the method.

The only minor weakness I would point is the lack of clarity in the presentation of the confidence intervals in the plots. It seems like the confidence intervals are applied around the non-smoothed curves which in practice makes them difficult to understand, I would suggest to add the confidence intervals around the smoothed curves and remove the raw curves from the plot to improve readability.

The main weakness I see, is the requirement of demonstrations in sparse reward tasks. In these kind of tasks, often times (not always), it might be difficult to have demonstrations that achieve a high return.

---

> ### Author Response · Authors · 2022-08-02
> **Response to Reviewer jXtd**
>
> We thank reviewer jXtd for the time spent reviewing and their valuable feedback. We have uploaded a new version of the manuscript and supplementary material in which we've addressed reviewer concerns, with text that has been added since the original submission updated to be blue.
>
> >I would suggest to add the confidence intervals around the smoothed curves and remove the raw curves from the plot to improve readability.
>
> Sorry for the confusion and thanks for the suggestion, this has been fixed!
>
> >The main weakness I see, is the requirement of demonstrations in sparse reward tasks. In these kind of tasks, often times (not always), it might be difficult to have demonstrations that achieve a high return.
>
> While demonstrations may not always be available, we argue that suboptimal demonstrations are often available for a number of tasks where dense reward specification is challenging. For example, a human may be able to slowly control a robot via a teleoperation interface (which can often be hard to use) without being able to provide a dense reward. Then, an RL algorithm can be used to smoothen and optimize the human provided trajectories. We would also like to point out that, as discussed in Section 2.1, this is a very common problem setting considered in a substantial body of prior work.
>
> >The main question I have is how important is the presence of the expert demonstrations in the replay buffer. Namely, have you measured the performance of the proposed algorithm only with the application of the MC targets in Eq. 5.4?
>
> In Section B.1 in the supplementary material we run a set of experiments without demonstration data. We find that in most environments no algorithms make any progress without demonstrations. In the one environment, sequential pushing, where it does make progress, MCAC performs similarly to default SAC. In future work, it would be interesting to evaluate in more environments where it is possible to make task progress without demonstrations to better answer this question.
>
> >Related to this, a study on the effect of the sub-optimality of the demonstrations should be performed. How much is the performance impacted from more sub-optimal demonstrations.
>
> In Figure (2a) in the supplement we perform a detailed study on the effect of demonstration quality by collecting demonstrations with different degrees of epsilon-greedy noise to the initial suboptimal demonstrator in the pointmass navigation domain and using them for MCAC. We find that MCAC does display some sensitivity to demonstration quality, though it does eventually make significant task progress for most values of epsilon. We have added a reference to these experiments in Section 6.4 for further clarity.

---

> > ### Comment · Reviewer_jXtd · 2022-08-06
> > **Reply to the responses**
> >
> > I thank the authors for their response and additional experiments.
> > I still think that my score of 6 is appropriate.

---

### Author Response · Authors · 2022-08-08
**Summary of Updates during Reviewer Discussion**

We thank all reviewers for their valuable comments and believe that the paper has been significantly improved as a result of their feedback. To summarize, the primary changes to the paper have been:

 * We added appendix B.1, B.5, B.6, B.7 where we evaluated MCAC without demonstration data, studied a variant on extending MCAC to infinite horizon problems, studied the scales of the Monte Carlo vs. TD estimators, and studied the interaction between AWAC and MCAC respectively. We believe these experiments allowed us to significantly deepen the intuition for MCAC’s strong performance in our results.
 * We made the discussion of experimental results in the paper more precise and added further clarification on the utility of MCAC for actor critic algorithms more broadly: it is a very simple to implement modification which appears to almost always help (and never significantly hurt) performance in our experiments when evaluated on two widely used algorithms (SAC and TD3), a complex return variant of SAC (GQE), and two SOTA RL from demonstrations algorithms (OEFD and AWAC) across 5 different continuous control tasks.
 * We updated the limitations section of the paper to reflect some of the reviewers’ concerns. Most importantly, because deep function approximators are difficult to reason about theoretically, we did not include theoretical guarantees for convergence or rigorous study of the MCAC objective’s effect on the bias-variance tradeoff for actor-critic algorithms. We agree with the reviewers that this would be an exciting direction to pursue in future work.

To the best of our knowledge, we have addressed all major reviewer concerns and would like to thank the reviewers and area chairs again for their careful consideration of our submission.

---

### Meta-Review · Area_Chair_aaxM · 2022-08-25

**Recommendation:** Accept
**Confidence:** Certain

**Metareview:**

This paper proposes two modifications of Actor-Critic algorithms for reinforcement learning with sparse reward: use demonstrations to seed the replay buffer, and combine TD Q target and Monte Carlo Q-target. All the reviewers agree that this paper has made good progress in an important research direction. The proposed method is simple-to-implement and seems to significantly boost the performance of the actor-critic methods. Most of the concerns in the original reviews were addressed through the rebuttal and discussions. However, one common concern that were raised by multiple reviewers, the lack of empirical/theoretical evidence of bias & variance reduction, remained unsolved. We understand that rigorous theoretical analysis for Deep RL algorithms, in general, is challenging. However, empirical analysis by conducting numerical experiments on simpler toy problems would still significantly improve the quality of this paper.

**Award:**

No

---

### Decision · Program_Chairs · 2022-09-14

Accept